# Photon production in top quark events at ATLAS and CMS

**Beatriz R. Lopes[1]\* on behalf of the ATLAS and CMS Collaborations**

**1** CERN, Geneva, Switzerland

\* beatriz.ribeiro.lopes@cern.ch

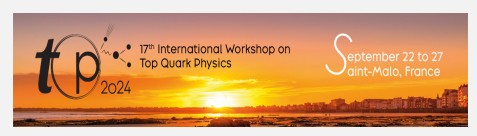

*The 17th International Workshop on
Top Quark Physics (TOP2024)
Saint-Malo, France, 22-27 September 2024*

## Abstract

**Top quark production in association with a photon offers a unique test ground for the standard model predictions, as it is sensitive to the top-photon coupling. These processes are rare when compared to standard top pair production, however the large amounts of data delivered by the LHC open the window to precise measurements. This talk covered the recent inclusive and differential measurements of top quark single and pair production in association with a photon, by the ATLAS and CMS Collaborations. Potential modifications to the top-photon couplings with respect to the standard model predictions are also explored using the standard model effective field theory.**

Copyright attribution to authors.
This work is a submission to SciPost Phys. Proc.
License information to appear upon publication.
Publication information to appear upon publication.

## 1  Introduction

Top quark measurements are a central part of the CERN LHC physics program, as it is the heaviest known elementary particle and has a large Yukawa coupling ($\sim$1). Processes with one or two top quarks and an associated hard and isolated photon, $t(\bar{t})\gamma$, although much rarer than simple $t(\bar{t})$ production, have relatively high cross sections when compared to the other top-boson processes. These processes are important backgrounds in several standard model (SM) measurements and searches for beyond the SM phenomena. Moreover, thanks to their sensitivity to the top-photon coupling, especially at high photon transverse momentum ($p_T$), they can also be used to search for new physics indirectly, namely using the SM effective field theory (EFT).

With the data collected at the LHC during Run 2, the $t\bar{t}\gamma$ process entered the precision era, with inclusive cross sections measured with a precision down to $\sim$4% [8]. Differential measurements have also been performed in several final states, by both ATLAS and CMS collaborations [6–8]. For the $tq\gamma$ process, observation (evidence) was reported by ATLAS (CMS) [11, 12], and inclusive cross section measurements have been performed. The $tW\gamma$

process, which is an important background to t̄tγ and interferes with it beyond the leading order, has only been measured summed with t̄tγ by ATLAS in eμ final states [**?**, 9].

## 2 Theoretical and experimental challenges

The t̄tγ process includes events with photons emitted from the initial state quarks, the top quarks, and the top quark decay products. These are experimentally indistinguishable, though their kinematics differ enough that it is possible to create phase spaces enriched in photons from a specific origin. For example, high $p_T$ photons originate mostly from the initial state or the top quarks. Nevertheless, in a t̄tγ analysis, one needs to model all possible photon origins. At leading order (LO) in quantum chromodynamics (QCD) this can be done relatively easily, however, at next-to-LO (NLO), simulating the full $2 \rightarrow 7$ process is not possible. One possible approach is to use a LO model and correct its cross section to the NLO value, and another is to use an NLO model which includes only photons from the initial state or off-shell top quarks (referred to as "t̄tγ production") stitched with a LO model filtered to include only photons from the on-shell tops or their decays (called "t̄tγ decay" in the following).

An additional challenge comes from the modelling of tWγ, which interferes with t̄tγ beyond the LO. This interference can in principle be removed using diagram removal (DR) or diagram subtraction (DS) strategies [5], however published results so far only use tWγ simulated samples at LO, where this interference is not present.

On the experimental side, selecting on the presence of a high $p_T$ isolated photon allows us to achieve very high signal purity (up to ∼80%). The main remaining background originates from t̄t events associated with nonprompt or "fake" photons. These can be electrons or jets misreconstructed as photons, as well as real photons that originate from the hadronisation process or from pileup events. The simulation doesn't always model these processes adequately and associated statistical uncertainties are typically large. Hence, data-driven methods are used to estimate their contribution.

In CMS, a strategy known as the ABCD method is typically used [13], where control regions enhanced in fake photons are built by inverting the selections on the charged isolation of the photon candidates and/or the width of the corresponding electromagnetic shower. The rate of fake photons is then determined in these regions and transported to the signal region. In the ATLAS analyses, a similar strategy is employed to estimate the contribution of photons from hadrons and pileup, while misreconstructed electrons are estimated from the fraction of electron–positron candidates from $Z \rightarrow ee$ decays that are reconstructed as $Z \rightarrow e\gamma$.

## 3 Inclusive cross section measurements of t̄tγ

The ATLAS and CMS collaborations measured the inclusive cross section of t̄tγ using data from the LHC Run 2 corresponding to around 140 fb$^{-1}$, in different fiducial phase spaces, in both dilepton and lepton+jets final states [6–8]. In all analyses, the events are selected based on single- and dileptonic triggers, and then required to have the appropriate number of leptons and jets, depending on the final state, and exactly one photon satisfying a number of quality criteria.

The most recent result comes from ATLAS [6] and focuses on both single lepton and dilepton channels. The "t̄tγ production" component is modelled at NLO in QCD while the "t̄tγ decay" component is modelled at LO in QCD. The total inclusive cross section is measured, and in an independent fit the t̄tγ production component is also measured, for the first time.

In each channel, a deep neural network (DNN) classifier is trained to separate the t̄tγ

production process from all other processes. The contribution from nonprompt photons is estimated from data as described in Sec. 2. A comparison between data and simulation for the outputs of the DNNs in the two channels is shown in Fig. 1 (left and centre).

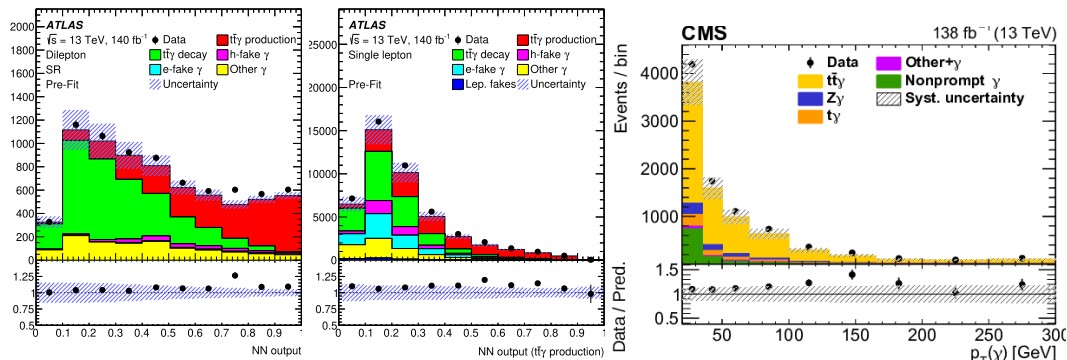

Figure 1: Scores of the NN ouputs in the signal region in the dilepton (left) and letpon+jets (centre) channels, in the analysis by ATLAS from Ref. [6]. Distribution of the photon $p_T$ (right) in the signal region, in the CMS analysis from Ref. [8].

The fit to the distributions in both channels simultaneously yields a cross section of

$$\sigma_{t\bar{t}\gamma \text{ (production)}} = 319 \pm 4 \text{ (stat)} \pm 15 \text{ (syst) fb (5\%)},$$

in agreement with the prediction of $296 \pm 30$ fb from MadGraph5_aMC@NLO. The measurement is dominated by the systematic uncertainties, mainly those on the modelling of $t\bar{t}\gamma$, the normalization of the $t\bar{t}\gamma$ decay component, as well as jet and b tagging uncertainties.

The CMS measurements focus on the lepton+jets [7] and dilepton [8] channels separately, and measure the fiducial cross sections for the total $t\bar{t}\gamma$ process (production+decay). In this case, the whole signal is modelled at LO in QCD with MadGraph5, scaled to NLO. Once again the nonprompt photon contribution is estimated with data-driven methods. Figure 1 (right) shows a comparison between data and simulation for photon $p_T$ in the dilepton channel. In the lepton+jets channel, a simultaneous fit to several signal and control regions is performed, while in the dilepton channel, one single signal region is fitted. The main systematic uncertainties affecting the measurement are those on signal modelling, background normalization, the estimation of nonprompt photons, and the luminosity. All results are compatible with theoretical calculations performed at NLO in QCD.

In Ref. [9], the $t\bar{t}\gamma$ and $tW\gamma$ processes are modelled at LO in QCD, and an inclusive fiducial cross section of the sum $t\bar{t}\gamma + tW\gamma$ is measured. The cross section is extracted from a fit to a signal region with one photon, and is in good agreement with dedicated fixed order calculations. Differential measurements are also performed, but they are not covered in this document, as they are partly superseded by the more recent results shown in Sec. 4. Dedicated $tW\gamma$ measurements with improved modelling do not exist yet and will be crucial for completing our understanding of these processes.

## 4 Differential cross section measurements of $t\bar{t}\gamma$

In Refs. [6–8], ATLAS and CMS also present differential measurements. All objects are defined at particle level, and the observables chosen for the differential measurement are the photon $p_T$ and $\eta$, and angular variables involving photons and jets or leptons (and for the CMS measurement in the dilepton channel, also jet kinematics). Normalised and absolute cross sections are measured for $t\bar{t}\gamma$ production+decay. The result from ATLAS includes also differential mea-

surements for the $t\bar{t}\gamma$ production process separately. Figure 2 shows two unfolded distributions for the absolute cross sections of the $t\bar{t}\gamma$ production process.

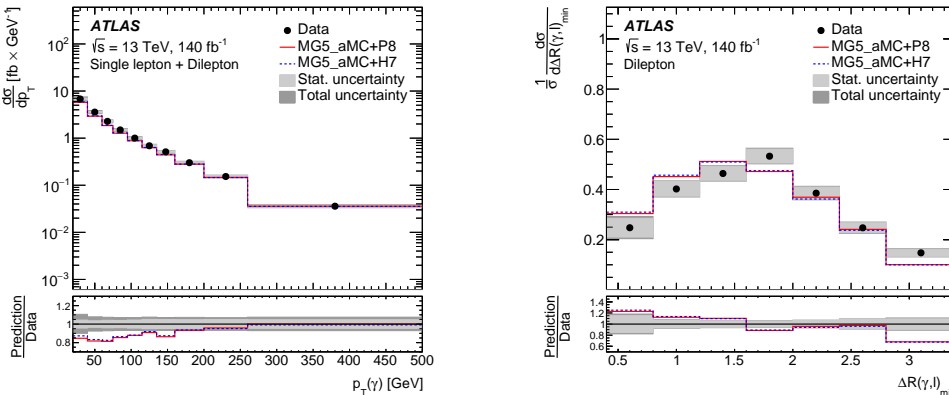

Figure 2: Absolute differential cross sections of $t\bar{t}\gamma$ production as a function of the photon $p_T$ in the lepton+jets and dilepton channels (left) and the angular distance between the photon and the closest lepton in the dilepton channel (right).

Figure 3 shows several unfolded distributions for the absolute cross sections of $t\bar{t}\gamma$ by CMS. The results from both collaborations show an overall agreement with the predictions, however the data present some clear trends with respect to the calculations, which most likely reflect the difficulties in modelling this process accurately.

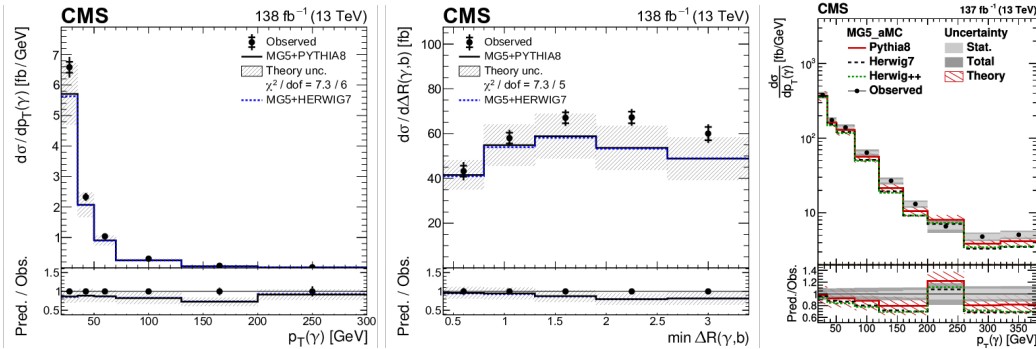

Figure 3: Absolute differential cross sections of $t\bar{t}\gamma$ as a function of the photon $p_T$ in the dilepton channel (left), the angular distance between the photon and the closest b jet in the dilepton channel (centre), and the photon $p_T$ in the lepton+jets channel (right) .

## 5    EFT interpretation

The photon $p_T$ distribution is sensitive to several EFT operators. Hence, a fit to this distribution allows extracting limits on the values of the corresponding Wilson coefficients. Both CMS and ATLAS perform such a fit and derive limits on the $c_{tZ}$ and $c_{tZ}^I$ coefficients. Operators modifying the $t\gamma$ coupling would also modify the tZ coupling; to exploit this, ATLAS performs the fit simultaneously to the photon $p_T$ in the $t\bar{t}\gamma$ region and to the Z boson $p_T$ in a $t\bar{t}Z$ region. The 2D contours of the upper limits obtained by CMS (ATLAS) on $c_{tZ}$ and $c_{tZ}^I$ using $t\bar{t}\gamma$ ($t\bar{t}\gamma$ and $t\bar{t}Z$) events are shown on the left (right) hand side of Fig. 4. In the ATLAS result, it is clear that the sensitivity is driven by $t\bar{t}\gamma$, even though the combination brings some additional constraints.

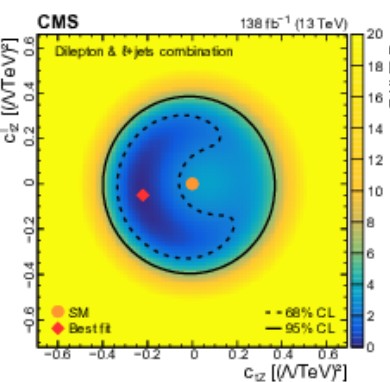 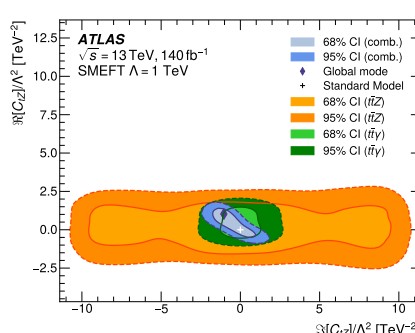

Figure 4: Left (right): Limits set by the CMS (ATLAS) analysis on the $c_{tZ}$ and $c_{tZ}^I$ coefficients. On the left, the red point indicates the measured best-fit value, while the dotted and solid lines show the observed 68 and 95% confidence levels, and the orange point shows the SM prediction. On the right, the best fit is shown by a blue diamond, while the contours for $t\bar{t}\gamma$ are the solid and dotted green lines, and the SM prediction is indicated by the white cross.

## 6 Top quark charge asymmetry in $t\bar{t}\gamma$ events

The top quark charge asymmetry ($A_c$) is an anisotropy in the angular distributions of the final-state top quark and antiquark, $A_C = \frac{\sigma_+ - \sigma_-}{\sigma_+ + \sigma_-}$, where $\sigma_{+(-)}$ is the cross section for positive (negative) values of $|y(t)| - |y(\bar{t})|$. For $t\bar{t}$ production, the SM prediction at NLO in QCD is of about 0.6%. In $t\bar{t}\gamma$ events, the charge asymmetry is potentially enhanced and expected to have the opposite sign compared to $t\bar{t}$, due to photon emission diagrams contributing to the interference, already at LO. The SM prediction at NLO varies between [-0.5%,-2%], depending on phase space choice.

This asymmetry is measured by ATLAS in Ref. [10], using a similar strategy to that of the differential cross section measurements. A NN is trained to separate $t\bar{t}\gamma$ production (signal) from the backgrounds, and the $A_c$ is extracted from a fit to $|y(t)| - |y(\bar{t})|$. The result is $A_c = -0.003 \pm 0.029$, in good agreement with the prediction. The measurement is limited by the statistical uncertainty.

## 7 Some words on $tq\gamma$

Measuring single top in association with a photon ($tq\gamma$) is an important additional input for EFT studies, and it is also an important background for $t\bar{t}\gamma$ in the lepton+jets channel. The first evidence for this process was reported by CMS in 2018, using 35.9 fb$^{-1}$ of data and only muon final states [11]. The process was later observed by ATLAS in 2023, using the full 140 fb$^{-1}$ of Run 2 [12]. In both analyses, the measured cross sections are 30-40% above the SM prediction, triggering the interest for further measurements of this process, especially differential ones.

## 8 Conclusion

This talk covered a number of recent results by ATLAS and CMS on top processes involving photon production. These processes, connecting the electroweak and strong sectors, provide a unique testing ground for the SM. The ATLAS and CMS collaborations have both measured the cross section of $t\bar{t}\gamma$ in the dilepton and lepton+jets channels, inclusively and differentially for a number of lepton, jet and photon observables. In the talk, I highlighted some recent improvements in the modelling strategy, which is one of the main challenges associated to this

 

measurement. Interpretations in the context of EFT by both ATLAS and CMS were also shown. Additionally, the measurements of the charge asymmetry using t$\bar{\text{t}}\gamma$ events and the observation of the tq$\gamma$ process were briefly introduced.

Some open tasks remain and new measurements can still be expected, exploiting the data collected in Run 2, and with the larger datasets being collected now in Run 3. For Run 3, the photon reconstruction and identification efficiencies have been computed, and some improvements with respect to Run 2 were achieved in CMS thanks to the use of more refined multivariate-based algorithms [14]. These improvements and the larger dataset are expected to open the way for even higher precision measurements of top-photon processes in the near future.

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
