# Peer review of "Photon production in top quark events at ATLAS and CMS"

_SciPost Physics Proceedings_

## Round 1 · Referee Report · Anonymous (Referee 1) · 2024-12-5

Report

Thanks so much for providing these nice proceedings. They are very well written and nicely summarize the talk. I only noticed that the ATLAS result was accepted and the numbers/plots slightly changed. Please update plots and numbers to the final version: https://arxiv.org/pdf/2403.09452 . Ref[6] should be adjusted accordingly. Otherwise, I have only minor corrections and adjustments to improve the document's clarity . Please find them below.

Textual comments:
p1, para1: Should be SM effective field theory (EFT)
to be consistent with the abstract

p1, para2: at the LHC in its Run 2 -> at the LHC during Run 2

p1, para2: suggest to add references to the individual results already here

p2, para2: phase spaces enriched in one photon origin with respect to the others. -> phase spaces enriched in photons from a specific origin.

p2, para2: The following is too unspecific: "at next-to-LO (NLO), it becomes computationally unfeasible"
I would suggest to make clear that this is about a 2->7 process that is not possible at NLO.

p2, para2: scale its cross section using a so-called k-factor -> correct its cross section to the NLO value

p2, para2: NLO model which includes only photons from the initial state or off-shell top quarks (referred to as "t¯tγ production") stitched with a LO model filtered to include only photons from the on-shell tops or their decays
Isn't it the other way around? Why are the on-shell top quarks in the NLO production?

p2, para3: Shouldn't one mention DR and DS?

p2, para4: Suggest to add Ref. Phys. Rev. D 44 (1991) 29 for the ABCD method.

p3, para3: Suggest to remove "using k-factors" as jargon. Scaled to NLO is good enough.

p3, para3: This needs some more info: two signal regions and six control regions. Suggest to quickly indicate how the regions are defined.
Or just say in several signal and control regions.

p5, para1: on phase space choices. -> on phase space choice.

Recommendation

Ask for minor revision

---

## Editorial Decision

editorial_decision: